Effect of hyperglycemia on the immune function of COVID-19 patients with type 2 diabetes mellitus: a retrospective study

Wang Ye 1
Yi Bo 1
Wang Shujun 2
Chen Xiaolin 1
Wen Zhongyuan 1 wenzytx@163.com
1 Department of Endocrinology, Renmin Hospital of Wuhan University , Wuhan, Hubei , China
2 Department of Obstetrics and Gynaecology, Renmin Hospital of Wuhan University , Wuhan, Hubei , China
Foti Daniela
Electronic publication date: 2022 Dec 16
Publication date: 2022
Volume: 10
Electronic Location ID: e14570
Received 2021 Jun 8; Accepted 2022 Nov 28
Copyright: © 2022 Wang et al.
Copyright year: 2022
Copyright holder: Wang et al.
License: This is an open access article distributed under the terms of the Creative Commons Attribution License, which permits unrestricted use, distribution, reproduction and adaptation in any medium and for any purpose provided that it is properly attributed. For attribution, the original author(s), title, publication source (PeerJ) and either DOI or URL of the article must be cited.
License URL: https://creativecommons.org/licenses/by/4.0/

Keywords: Corona Virus Disease 2019, Type 2 diabetes mellitus, Hyperglycemia, Immune function

Funding: Zhejiang University Special Scientific Research 2020XGZX015 National Natural Science Foundation of China 81900749 Nature Science Foundation of Hubei province 2018CFB150 The work was supported by the Zhejiang University special scientific research fund for COVID-19 prevent and control (grant number 2020XGZX015), the National Natural Science Foundation of China (grant number 81900749), and the Nature Science Foundation of Hubei province (grant number 2018CFB150). The funders had no role in study design, data collection and analysis, decision to publish, or preparation of the manuscript.

==============================
Purpose

To analyze the clinical characteristics and immune function parameters and to explore the effect of hyperglycemia on the immune function in patients with Corona Virus Disease 2019 (COVID-19) with type 2 diabetes mellitus (T2DM).

Methods

This retrospective study included patients with COVID-19 with T2DM hospitalized in Renmin Hospital of Wuhan University between January 31, 2020, and February 10, 2020. The clinical data were collected and patients were divided into a well-controlled group (blood glucose 3.9–10.0 mmol/L) and a poorly-controlled group (blood glucose >10.0 mmol/L). The differences in routine blood tests, peripheral lymphocyte subsets, humoral immune components, C-reactive protein (CRP) level, and cytokines were compared, and the correlation between blood glucose and immune parameters as well as the severity of the disease was analyzed.

Results

A total of 65 patients with COVID-19 and T2DM were included in the final analysis. Compared with the well-controlled group, patients in the poorly-controlled group had decreased lymphocytes, CD16+ 56+ NK cells, CD3+ T cells, CD8+ T cells and increased neutrophil percentage, IL-6 levels, CRP levels and serum concentration of IgA. Blood glucose was inversely correlated with CD16+ 56+ NK cells, CD3+ T cells, CD4+ T cells, and CD8+ T cells and positively correlated with IL-6 and CRP levels. There was a positive correlation between blood glucose and the severity of the COVID-19.

Conclusion

Hyperglycemia will aggravate the immune dysfunction of COVID-19 patients with T2DM and affect the severity of COVID-19.

Introduction

Corona Virus Disease 2019 (COVID-19) is caused by an emerging infection with a highly infectious coronavirus, SARS-CoV-2 (Chen et al., 2020). According to the official data of World Health Organization (WHO), a total of 162,177,376 were confirmed to have COVID-19 worldwide as of May 16, 2021 (World Health Organization, 2021), which causing enormous damage to the global economy and medical treatment. Relevant studies have confirmed that patients with diabetes are not only more susceptible to COVID-19, they are also more likely to have a poor prognosis of COVID-19 (Liu et al., 2020a, 2020b). This can be associated to the following reasons: (1) the impairments of the innate and humoral immune systems caused by metabolic dysfunction in patients with diabetes makes them susceptible to infectious diseases (Wang et al., 2020d); (2) binding of the viral spike (S) proteins to cellular receptors and S protein priming by host cell proteases are the key steps for coronavirus to enter cells, previous study demonstrated that SARS-CoV-2 uses the SARS-CoV receptor angiotensin-converting enzyme 2 (ACE2) for entry and the serine protease TMPRSS2 for S protein priming (Hoffmann et al., 2020; Matarese et al., 2020). The upregulation of ACE2 expression in DM cardiomyocytes, along with non-enzymatic glycation, could increase the susceptibility to COVID-19 infection in DM patients by favoring the cellular entry of SARS-CoV2, which lead to a worse prognosis (D’Onofrio et al., 2021). Compared to non-DM, it was also found elevated TMPRSS2 protein expression in COVID-19 myocardial tissue from DM patients.

COVID-19 patients often have severe immune dysfunction. If this dysfunction persists, it will cause a cytokine storm and eventually lead to deterioration or even death (Ye, Wang & Mao, 2020). Blood glucose level affects the risk of death as well as the prognosis of diabetic patients with SARS-CoV-2 infection (Zhu et al., 2020). The association between hyperglycemia and immune function in COVID-19 patients with type 2 diabetes mellitus (T2DM) remains unclear. Therefore, we retrospectively analyzed the clinical data of COVID-19 patients admitted to Renmin Hospital of Wuhan University from January 31 to February 10, 2020, to figure out the effect of hyperglycemia on the immune function in COVID-19 patients with T2DM.

Materials and Methods

Study subjects

The study included 2,120 patients diagnosed with COVID-19 admitted to Renmin Hospital of Wuhan University between January 31, 2020, and February 10, 2020. Those patients previously diagnosed with T2DM were extracted. This study was approved by the Clinical Research Ethics Commission of Renmin Hospital of Wuhan University(WDRY2020-K051).

Diagnostic criteria

The Diagnosis of COVID-19 was determined according to the Guidelines of the Diagnosis and Treatment of New Coronavirus Pneumonia (Edition 5) published by the National Health Commission of China (National Health Commission of the People’s Republic of China, 2020). Diagnostic criteria included epidemiological history, clinical presentation of fever and respiratory symptoms, imaging features of pneumonia, and etiological evidence of positive SARS-CoV-2 nucleic acid tested through real-time fluorescent reverse transcription-polymerase chain reaction (RT-PCR) on nasopharyngeal swab specimens. The severity of COVID-19 was also determined according to the Guidelines as follows: a) Mild cases. Mild clinical symptoms but no imaging findings of pneumonia.

b) Common type. Fever and other respiratory symptoms, imaging findings of pneumonia on chest computed tomography (CT)

c) Severe cases. Meet at least one of the following criteria: (i) respiratory distress, respiratory rate ≥30 beats/min, (ii) hypoxemia, finger oxygen saturation ≤93% at rest; (iii) partial pressure of arterial oxygen (PaO2)/fractional inspired oxygen (FiO2) ≤300 mmHg (1 mmHg = 0.133 kPa).

d) Critical cases. Meet at least one of the following criteria: (i) respiratory failure and requirement of mechanical ventilation, (ii) shock, (iii) other organ failure requiring ICU monitoring and treatment occurred.

The diagnosis of diabetes was based on the 1999 World Health Organization diagnostic criteria.

Inclusion criteria: Patients diagnosed with COVID-19 with a history of type 2 diabetes.

Exclusion criteria: (1) Type 1 diabetes and specific type diabetes; (2) Pregnancy; (3) History of severe kidney and liver disease; (4) History of autoimmune disease; (5) History of tumor; (6) History of chronic respiratory disease; (7) History of glucocorticoid treatment.

Data collection

Demographics, clinical information, laboratory results, and imaging findings were collected during the participants’ hospitalization. Laboratory results included first examination of routine blood tests, blood glucose level (usually a random blood glucose), peripheral lymphocyte subsets (CD16+56+ NK cells, CD19+ B cells, CD3+ T cells, CD4+ T cells, and CD8+ T cells), humoral immune components (Immunoglobulins [Ig] G, IgM, IgA, IgE, complement proteins [C]3 and C4), C-reactive protein (CRP) level, and cytokines (including Interleukin [IL]-2, IL-4, IL-6, IL-10) of the patient after admission. Chest CT performed by two attending physician with 3 years of experience in CT reading. They were blinded to study cohorts and study protocol. The first chest CT images results were collected after admission. All data were independently reviewed and entered into a computer database by two analysts (Y.W. and B.Y.).

SARS-CoV-2 testing for all patients was conducted in the clinical laboratory of Renmin Hospital of Wuhan University. According to the instructions of the kit, nucleic acid was extracted using the viral nucleic acid kit after the clinical samples (nasopharyngeal swab specimens) were obtained. ORF1ab gene (nCovORF1ab) and N gene (nCoV-NP) were detected by the method of real-time RT-PCR using the 2019-nCoV assay kit (Bioperectus) according to the manufacturer’s instructions. A laboratory-confirmed infection was considered if both nCovORF1ab and nCoV-NP testing results were positive.

Statistical analysis

Descriptive statistics were performed for all study variables. SPSS 26.0 (IBM) software was applied to analyze the data. Fisher exact test or X2 test was employed for comparing the results of all categorical variables. Measurement data of continuous variables were first tested for normal distribution. Data meeting normal distribution were expressed as mean ± standard deviation ( X¯ ± s), and the comparison was performed using t-test; data not meeting normal distribution were expressed as median and interquartile range (IQR) values, and the comparison was performed using the Mann-Whitney U test. The Pearson or Spearman method was selected for correlation analysis according to whether the data conformed to the normal distribution. Multivariate logistic regression analysis was performed based on the severity of COVID-19. Statistical figures were prepared using Prism 5 (Graphpad). In all statistical analyses, the significance level (α) was 0.05. P < 0.05 was considered to indicate a significant difference.

Results

Clinical characteristics of patients with COVID-19 with T2DM

Figure 1 depicts a flowchart for patient screening. Through collecting the clinical data of 2,120 confirmed COVID-19 patients admitted to Renmin Hospital of Wuhan University from January 31 to February 10, 2020, a total of 107 patients with previously confirmed diabetes were first selected. Among them, four patients with gestational diabetes mellitus, one patient with latent autoimmune diabetes in adults (LADA), one patient with drug-induced diabetes mellitus, and three patients with tumor were subsequently excluded. Besides, 33 patients with missing clinical data (including laboratory and imaging results) were also screened out. Finally, 65 patients with COVID-19 and pre-existing T2DM were included in this study. They were divided into a well-controlled group (blood glucose: 3.9–10.0 mmol/L) and a poorly-controlled group (blood glucose >10.0 mmol/L) according to the admission blood glucose levels.

Figure 1 Screening flowchart.

Through collecting the clinical data of 2,120 confirmed COVID-19 patients admitted to Renmin Hospital of Wuhan University from January 31 to February 10, 2020, a total of 107 patients with previously confirmed diabetes were first selected. According to the exclusion criteria, nine patients were subsequently excluded. In addition, 33 patients with missing clinical data were also screened out. Finally, 65 patients with COVID-19 and pre-existing T2DM were included in this study. They were divided into a well-controlled group (blood glucose: 3.9–10.0 mmol/L) and a poorly-controlled group (blood glucose >10.0 mmol/L) according to the admission blood glucose levels.

The mean age of the 65 patients was 66 ± 11 years, including 35 males and 30 females. The onset symptoms of the patients were analyzed. The most common symptoms were fever (51 patients (78.5%)). Less common symptoms were cough (39 patients (60.0%)), fatigue (31 patients (47.7%)), and dyspnea (29 patients (44.6%)), chest tightness (15 patients (23.1%)), diarrhea (12 patients (18.5%)), myalgia (six patients (9.2%)) and anorexia (five patients (7.7%)). The most common comorbidity of these patients was hypertension (41 patients (63.1%)). Chest radiography and computed tomography (CT) findings revealed bilateral pulmonary lesions in all 65 patients. A total of 46 (83.6%) patients were classified as having common COVID-19, 16 patients (24.6%) were classified as having severe COVID-19, and three patients (4.6%) were classified as having critical COVID-19. Further analysis revealed that the patients with coronary heart disease in the well-controlled group were more than that in the poorly-controlled group (11 (30.6%) vs. 2 (6.7%), P = 0.034). The information of glucose-lowering medicine, antihypertensive medicine and COVID-19 medication of 65 patients during hospitalization was listed in Table 1 (shown as case (%)). No statistical differences were found between the two groups for all other baseline characteristics (Table 1).

Table 1 Demographics, baseline characteristics and radiographic findings of 65 patients with COVID-19 with T2DM.

Clinical characteristics on admission	Normal range	COVID-19 with T2DM (N = 65)	Well-controlled
(n = 35)	Poorly-controlled
(n = 30)	P value	
Age	–	66 ± 11	67 ± 9	65 ± 13	0.536	
Male/Female	–	35/30	22/13	13/17	0.115	
SBP (mmHg)	–	131 ± 17	132 ± 18	130 ±17	0.750	
DBP (mmHg)	–	77 ± 12	77 ± 13	76 ± 11	0.592	
Major symptoms	–					
Fever, n (%)	–	51 (78.5)	29 (82.9)	22 (73.3)	0.352	
Cough, n (%)	–	39 (60.0)	21 (60.0)	18 (60.0)	>0.99	
Fatigue, n (%)	–	31 (47.7)	17 (48.6)	14 (46.7)	0.878	
Dyspnoea, n (%)	–	29 (44.6)	16 (45.7)	13 (43.3)	0.944	
Chest tightness, n (%)	–	15 (23.1)	8 (22.9)	7 (23.3)	0.964	
Diarrhoea, n (%)	–	12 (18.5)	5 (14.3)	7 (23.3)	0.349	
Myalgia, n (%)	–	6 (9.2)	2 (5.7)	4 (13.3)	0.530	
Anorexia, n (%)	–	5 (7.7)	2 (5.7)	3 (10.0)	0.857	
Comorbidities on admission	–					
Hypertension, n (%)	–	41 (63.1)	18 (60.0)	23 (65.7)	0.634	
Coronary heart disease, n (%)	–	13 (19.7)	11 (30.6)	2 (6.7)	0.034	
Hyperuricemia, n (%)	–	2 (3.1)	1 (2.9)	1 (3.3)	>0.99	
Hyperlipidemia, n (%)	–	2 (3.1)	1 (2.9)	1 (3.3)	>0.99	
Kidney disease, n (%)	–	2 (3.1)	1 (2.9)	1 (3.3)	>0.99	
Liver disease, n (%)	–	3 (4.6)	1 (2.9)	2 (6.7)	0.891	
Glucose-lowering
Medication						
Insulin n (%)	–	42 (64.6%)	17 (48.6)	25 (83.3)	0.003	
Alpha-glucosidase inhibitor n (%)	–	37 (56.9%)	18 (51.4%)	19 (63.3%)	0.334	
Metformin n (%)	–	25 (38.5%)	12 (34.3%)	13 (43.3%)	0.455	
Repaglinide n (%)	–	9 (13.8%)	6 (17.1%)	3 (10.0%)	0.406	
Glimepiride n (%)	–	8 (12.3%)	6 (17.1%)	2 (6.7%)	0.200	
Gliclazide n (%)	–	7 (10.8%)	2 (5.7%)	5 (16.7%)	0.156	
Antihypertensive medication						
CCB n (%)	–	15 (23.1)	9 (50.0)	6 (69.6)	0.334	
ARB/ACEI n (%)	–	13 (20.0)	8 (44.4)	5 (21.7)	0.179	
β-R blockers n (%)	–	9 (13.8)	4 (22.2)	5(21.7)	1.000	
COVID-19 Medication	–					
Antiviral	–	63 (96.9%)	34 (97.1%)	29 (96.7%)	1.000	
Traditional Chinese medicine	–	59 (90.8%)	30 (85.7%)	29 (96.7%)	0.275	
Antibiotics	–	46 (70.8%)	25 (71.4%)	21 (70.0%)	0.900	
Intravenous-	–	33 (50.8%)	17 (48.6%)	16 (53.3%)	0.702	
Immuneglobulin						
Glucocorticoids	–	32 (49.2%)	16 (45.7%)	16 (53.3%)	0.540	
Thymosin	–	15 (23.1%)	8 (22.9%)	7 (23.3%)	0.964	
Recombinant human interferon α-2α	–	7 (10.8%)	2 (5.7%)	5 (16.7%)	0.308	
Chest radiography and computed tomography findings	–					
Bilateral lesions, n (%)	–	65 (100)	35 (100)	30 (100)	—	
Severity of COVID-19	–					
Common cases, n (%)	–	46 (83.6)	28 (80)	18 (90)	0.335	
Severe cases, n (%)	–	16 (24.6)	6 (17.1)	10 (33.3)	0.131	
Critical cases, n (%)	–	3 (4.6)	1 (2.9)	2 (6.7)	0.891	
Library Results	–					
Glucose (mmol/L)	3.9–6.1	9.70 (7.13–15.70)	7.18 (5.67–8.90)	16.61 (12.1–20.23)	0.000	
Leukocyte count (×109/L)	3.5–9.5	6.45 (5.26–7.95)	5.93 (5.26–7.44)	6.59 (5.29–8.70)	0.295	
Low or normal leukocyte count n (%)	–	55 (84.6)	30 (85.7)	25 (83.3)	0.791	
Lymphocyte count
(×109/L)	1.1–3.2	0.99 (0.64–1.33)	1.01 (0.84–1.42)	0.80 (0.54–1.17)	0.028	
Lymphocyte ratio (%)	20–50	14.6 (9.8–23.1)	17.9 (10.4–27.9)	12.35 (6.80–19.03)	0.019	
Lymphopenia count n (%)	–	40 (61.5)	19 (54.3)	21 (70.0)	0.194	
Neutrophils count
(×109/L)	1.8–6.3	4.86 (3.40–6.58)	4.40 (2.91–6.22)	5.16 (3.84–7.41)	0.107	
Neutrophils ratio (%)	40–75	76.0 (66.2–85.2)	71.80 (61.00–80.30)	78.35 (71.78–87.48)	0.022	
Abnormal Neutrophils count (<1.8 × 109 or >6.3 × 109/L) n (%)	–	17 (26.2)	8 (22.9)	9 (30.0)	0.514	
Red blood cell count
(×1012/L)	3.8–5.1	4.12 (3.71–4.32)	4.12 (3.74–4.44)	4.08 (3.58–4.34)	0.617	
Hemoglobin content (g/L)	115–150	126 (117–134)	130 (121–135)	121 (105.5–129.0)	0.040	
Platelet count (×1012/L)	125–350	214 (176–285)	214 (169–292)	209.5 (174.25–259.5)	0.519	
CD16+56+ NK cells (cells/μL)	84–724	115.0 (63.5–181.0)	143.0 (97.0–203.0)	83.5 (42.3–163.0)	0.005	
CD19+ B cells (cells/μL)	80–616	143.0 (94.0–217.0)	146 (119–187)	131.0 (84.5–243.5)	0.732	
CD3+ T cells (cells/μL)	723–2737	524.0 (304.0–862.5)	639.0 (402.0–1,020.0)	442.0 (247.8–777.8)	0.042	
CD4+ T cells (cells/μL)	404–1612	366.0 (202.0–617.5)	420 (288–648)	299 (170–542)	0.107	
CD8+ T cells (cells/μL)	220–1129	138.0 (83.5–245.5)	176.0 (109.0–295.0)	119.0 (71.5–199.8)	0.029	
C3 (g/L)	0.7–1.4	1.00 (0.87–1.14)	0.98 (0.86–1.07)	1.02 (0.88–1.19)	0.403	
C4 (g/L)	0.1–0.4	0.23 (0.18–0.33)	0.22 (0.15–0.35)	0.24 (0.19–0.31)	0.698	
IgA (g/L)	1.0–4.2	2.93 (2.39–3.93)	2.55 (2.20–3.53)	3.42 (2.85–4.20)	0.029	
IgE (IU/ML)	<100	53.9 (18.3–114.5)	39.6 (18.3–94.8)	70.2 (20.6–118.3)	0.242	
IgG (g/L)	8.6–17.4	13.0 (10.9–14.8)	12.1 (10.0–14.4)	13.4 (11.6–15.3)	0.244	
IgM (g/L)	0.3–2.2	0.84 (0.67–1.09)	0.84 (0.72–1.23)	0.83 (0.62–0.99)	0.206	
IL-2 (pg/mL)	<11.4	3.86 (3.55–4.34)	3.85 (3.53–4.38)	3.99 (3.56–4.28)	0.885	
IL-4 (pg/mL)	<12.9	3.32 (3.04–3.81)	3.27 (3.01–3.81)	3.32 (3.10–3.82)	0.422	
IL-6 (pg/mL)	<20	6.98 (3.93–13.91)	6.11 (3.55–10.22)	10.98 (4.40–22.97)	0.030	
IL-10 (pg/mL)	<5.9	5.84 (5.17–7.09)	5.77 (5.02–6.85)	5.84 (5.25–7.35)	0.576	
CRP (mg/L)	0–6	21.9 (5.0–76.2)	21.9 (5.0–66.2)	45.5 (19.5–82.1)	0.034	
Note:

Statistical significance was determined at P < 0.05. Abbreviation: SBP, Systolic blood pressure; DBP, Diastolic blood pressure; IL, Interleukin; CRP, C-reactive protein; CCB, Calcium channel blockers; ARB, Angiotensin II-R blockers; ACEI, Angiotensin Converting Enzyme Inhibitors.

Laboratory findings

Routine blood analysis

A total of of 65 patients received routine blood analysis. The results showed that the median white blood cell count was in the normal range (median [IQR], 6.45 [5.26–7.95] × 109/L), and 55 patients with normal or reduced white blood cell count. The median lymphocyte count and percentage were both decreased (median [IQR], 0.99 [0.64–1.33] × 109/L; 14.6 [9.8–23.1]%), and decreased lymphocyte count was detected in 40 patients (61.5%). The median neutrophil count was in the normal range (median [IQR], 4.86 [3.40–6.58] ×109/L), but the median neutrophil percentage was increased (median [IQR], 76.0 [66.2–85.2]%). Abnormal neutrophils count was detected in17 patients (26.2%). Further analysis revealed that compared with the well-controlled group, the lymphocyte count and lymphocyte percentage decreased more significantly (median [IQR], 0.80 [0.54–1.17] × 109/L vs. 1.01 [0.84–1.42] × 109/L, P = 0.028; 12.35 [6.80–19.03]% vs. 17.9 [10.4–27.9]%, P = 0.019), the neutrophil percentage increased significantly (median [IQR], 78.35 [71.78–87.48]% vs. 71.80 [61.00-80.30]%, P = 0.022), and the hemoglobin concentration decreased (median [IQR], 121 [105.5–129.0] g/L vs. 130 [121–135] g/L, P = 0.040) in the poorly-controlled group (Table 1).

Peripheral lymphocyte subsets analysis

Lymphocyte subsets analyses were carried out in the 65 patients. We found the median count of the CD3+ T cells, CD4+ T cells, and CD8+ T cells were all below the lower limit of the normal range (shown in Table 1). Compared with the well-controlled group, patients in the poorly-controlled group had significantly lower count of CD16+56+ NK cells (median [IQR], 83.5 [42.3–163.0]/μL vs. 143.0 [97.0–203.0]/μL, P = 0.005), CD3+ T cells (median [IQR], 442.0 [247.8–777.8]/μL vs. 639.0 [402.0–1020.0]/μL, P = 0.042), CD8+ T cells (median [IQR], 119.0 [71.5–199.8]/μL vs. 176.0 [109.0–295.0]/μL, P = 0.029). Although there was no statistical significance, poorly-controlled group have lower count of CD4+ T cells and CD19+ B cells (Table 1 and Fig. 2).

Figure 2 The Peripheral lymphocyte subsets between well-controlled group and poorly-controlled group.

(A) CD16+56+ NK cells. (B) CD19+ B cells. (C) CD3+ T cells. (D) CD4+ T cells. (E) CD8+ T cells. NS: no significance; *P < 0.05 ; **P < 0.01.

Humoral immune components analysis

The median serum concentration of humoral immune components including C3, C4, IgA, IgE, IgG, and IgM were in the normal range in all 65 patients. The median serum concentration of IgA in the poorly-controlled group was significantly higher than that in the well-controlled group (median [IQR], 3.42 [2.85–4.20] g/L vs. 2.55 [2.20–3.53] g/L, P = 0.029). Although there was no statistical significance, poorly-controlled group have higher serum concentration of C3, C4, IgE, and IgG (Table 1).

CRP and cytokines

The median levels of serum IL-2, IL-4, IL-6, and IL-10 were in the normal range in all 65 patients. The median level of CRP, an inflammatory marker, was higher than higher limit of the normal range (median [IQR], 21.9 [5.0–76.2] mg/L). The median levels of IL-6 and CRP in the poorly-controlled group were higher than those in the well-controlled group (median [IQR], 10.98 [4.40–22.97] pg/mL vs. 6.11 [3.55–10.22] pg/mL, P = 0.030; 45.5 [19.5–82.1] mg/L vs. 21.9 [5.0–66.2] mg/L, P = 0.034). There was no statistical difference in IL-2, IL-4, and IL-10 levels between the two groups (Table 1).

Correlation between blood glucose and immune parameters

In order to confirm the effect of blood glucose on immune function in patients with COVID-19 with T2DM, the correlation between blood glucose and immune parameters was further analyzed in the 65 patients. Blood glucose was found to be negatively correlated with the count of CD16+56+ NK cells, CD3+ T cells, CD4+ T cells, and CD8+ T cells (r = −0.320, P = 0.009; r = −0.384, P = 0.002; r = −0.338, P = 0.006; r = −0.393, P = 0.001, respectively). Besides, the blood glucose was also positively correlated with IL-6 and CRP levels (r = 0.322, P = 0.009; r = 0.329, P = 0.007, respectively) (Fig. 3). There was no correlation between blood glucose and CD19+ B cell count, IL-2, IL-4, IL-10, as well as humoral immune components.

Figure 3 Correlation between blood glucose and immune parameters.

(A) Blood glucose and CD16+ 56+ NK cells. (B) Blood glucose and CD3+ T cells. (C) Blood glucose and CD4+ T cells. (D) Blood glucose and CD8+ T cells. (E) Blood glucose and IL-6 levels. (F) Blood glucose and CRP levels.

Factors predicting the severity of COVID-19 patients with T2DM

We first performed a correlation analysis between blood glucose and severity of COVID-19 in the 65 patients. The result showed a positive correlation between blood glucose and severity of COVID-19 (r = 0.257, P = 0.04). Then we performed binary logistic regression analysis to evaluate the correlation between severity of COVID-19 and other independent variables (age, lymphocyte count, median neutrophil count, CD16+56+ NK cells, CD3+ T cells, CD4+ T cells, CD8+ T cells, IL-2, IL-4, IL-6 and IL-10 level), the result showed that CD3+ T cells, CD4+ T cells, and CD8+ T cells were risk factors for the severity of COVID-19 patients with T2DM (Table 2).

Table 2 Binary logistic regression analysis with the clinical classification as the dependent variable in 65 patients with COVID-19 with T2DM.

	B	S.E	Wals	Sig.	Exp (B)	95% CI	
CD3+ T cells	−0.095	0.039	5.903	0.015	0.910	[0.843–0.982]	
CD4+ T cells	0.090	0.038	5.590	0.018	1.094	[1.016–1.179]	
CD8+ T cells	0.085	0.039	4.753	0.029	1.089	[1.009–1.175]	
Note:

Statistical significance was determined at P < 0.05.

Discussion

Elderly COVID-19 patients are mostly complicated with underlying diseases such as diabetes, hypertension, and coronary heart disease (Chen et al., 2020; Wang et al., 2020a; Zhou et al., 2020). Chronic hyperglycemia is known to downregulate ACE2 expression making the cells vulnerable to the inflammatory and damaging effect of the virus (Bornstein et al., 2020). This study demonstrated that hyperglycemia could aggravate the immune dysfunction in COVID-19 patients with known type-2 diabetes. Meanwhile, chronic hyperglycemia is associated with the severity of COVID-19.

In the present study, we discovered that fever, cough, fatigue, and dyspnea were the common clinical symptoms in the 65 patients with COVID-19 with T2DM, which was consistent with previous studies in COVID-19 patients with diabetes (Wang et al., 2020c). Chest radiography and computed tomography showed bilateral pulmonary lesions in all 65 patients, indicating that patients with T2DM had severe pulmonary lesions after infection with SARS-CoV-2. Lymphocytes can be divided into four subsets: helper T cells (Th, CD3+CD4+ expression), cytotoxic T cells (Tc, CD3+CD8+ expression), natural killer (NK) cells (CD16+CD56+ expression), and B cells (CD19+ expression). These cells are involved in cellular and humoral immunity against viral infections and play an important role in maintaining immune system function (Wang et al., 2020b). Lymphocytes can recognize antigens, secrete a variety of inflammatory cytokines, and increase the immune capacity of the body, while Tc cells can directly kill target antigen cells, such as cells infected with microorganisms. CD8+T cells play a key role in mediating virus clearance in acute respiratory infections (Mittrücker, Visekruna & Huber, 2014; Van Den Brand, Smits & Haagmans, 2015). The infection of SARS-CoV-2 can lead to lymphocytopenia and imbalance of lymphocyte subsets, such as decreased CD4+ T cells, CD8+ T cells, B cells, and NK cells (Guan et al., 2020; Wang et al., 2020b; Zhang et al., 2020; Xu et al., 2020). Research have shown that SARS-CoV-2 has destroyed the antiviral immune capacity by depleting T lymphocytes in the early stage of infection (Zheng et al., 2020). In this study, we also showed that CD3+ T cells, CD4+ T cells, and CD8+ T cells were significantly reduced in patients with COVID-19 with T2DM, suggesting that these patients had significant dysregulation in lymphocyte, leading to impaired function of cellular immune and decreased viral clearance. Hyperglycemia can lead to alterations in lymphocytes, resulting in diminished T cell immune response and impaired function of cytokine secretion (Daryabor et al., 2020). We found lymphocytopenia was more severe in the poorly-controlled group than well-controlled group. The count of CD3+ T cells, CD8+ T cells and CD16+CD56+ NK cells were significantly decreased in the poorly-controlled group and blood glucose was negatively correlated with count of CD16+CD56+ NK cells, CD3+ T cells, CD4+ T cells, and CD8+ T cells. Those results further confirmed that hyperglycemia exacerbate the impairment of cellular immune function in COVID-19 patients with T2DM.

In previous study, it was found elevated serum level of IL-6 in COVID-19 patients with diabetes compared with those without diabetes (Zheng et al., 2021). We found that the median level of CRP and IL-6 in the poorly-controlled group was significantly higher than that in the well-controlled group. We also found that blood glucose level was positively correlated with the serum concentration of IL-6 and CRP level, suggesting that hyperglycemia aggravated the systemic inflammatory response caused by SARS-CoV-2.

The high risk of progression to severe-stage COVID-19 in patients with diabetes is likely because of hyperglycemic conditions that cause immune dysfunction including impaired neutrophil function, antioxidant system function, and humoral immunity (Chang et al., 2020). We showed that the increased neutrophil percentage was more significant in the poorly-controlled group. It may contribute to secondary bacterial infection in COVID-19 patients with hyperglycemia. It was proved significant humoral immune dysfunction in COVID-19 patients with T2DM because we did not find elevated serum concentrations of C3, C4, IgA, IgE, IgG, and IgM. The mucosal response system of the respiratory tract is the first line of defense against viral respiratory tract infections (Sato & Kiyono, 2012). It has been discovered that the specific IgA antibodies of SARS-CoV-2 may play an important role in providing protective mucosal responses (Ejemel et al., 2020). Recent reports indicated that early SARS-CoV-2-specific humoral responses were dominated by IgA antibodies. Furthermore,previous study showed that IgA was a regulator of immune hyperactivation (Olas et al., 2005). Yu et al. (2020) found that serum IgA levels were significantly higher in severe COVID-19 patients and high levels of serum IgA may have adverse effects in severe patients. Interestingly, we found that patients in the poorly-controlled group had relatively higher serum IgA level than those in the well-controlled group. Whether the elevated serum IgA levels was a result of defense response in COVID-19 patients with hyperglycemia or not needs to be further studied.

Sardu et al. (2020) found that a decrease in glucose levels between baseline and 24 h was associated with a lower rate of progression to severe disease and death at 20 days in both nondiabetic and diabetic hyperglycaemic patients with COVID-19. Our data indicated that blood glucose was positively correlated with the severity of the condition of 65 patients, which further suggested that hyperglycemia can affect the condition of COVID-19 patients with T2DM. However, other factors were not found to be associated with the severity, that might be related to the fact that most patients had mild conditions and the sample size was small.

Conclusion

In summary, we conclude that COVID-19 patients with T2DM have obvious immune dysfunction, and poor control of blood glucose (hyperglycemia) aggravates the disturbance of immune response to SARS-CoV-2 in T2DM patients and affects the severity of COVID-19. This study also suggests that good in-patient glycemic control can reduce the further aggravation of immune disorders in COVID-19 patients with T2DM, which is very important in the comprehensive treatment of COVID-19.

This study has several limitations. First, the sample size was small, due to the clinical data of a large number of COVID-19 patients with T2DM were missing or incomplete, and patients with acute hyperglycemia were not included in the study. Second, the lack of detection results of glycosylated hemoglobin A1C, insulin level, TNF-α, IFN, IL-1β, and other cytokines limited the deeper assessment of glycemia and immune dysfunction in COVID-19 patients with T2DM. Third, the first laboratory data was selected for this study, which could not accurately reflect the continuous dynamic changes of immune function parameters. Fourth, more in vitro evidence is needed to explore the significance of IgA levels for COVID-19 patients with T2DM.

Supplemental Information

Supplemental Information 1 Raw data on the general condition, drugs, laboratory tests, and correlation analysis of enrolled cases.

The PZF file needs to be opened with GraphPad Prism.

Click here for additional data file.

We acknowledge all the medical staff who worked hard for patients during the COVID-19 epidemic.

Additional Information and Declarations

Competing Interests

Author Contributions

Human Ethics

Data Availability

The authors declare that they have no competing interests.

Ye Wang conceived and designed the experiments, performed the experiments, analyzed the data, prepared figures and/or tables, authored or reviewed drafts of the article, and approved the final draft.

Bo Yi conceived and designed the experiments, authored or reviewed drafts of the article, and approved the final draft.

Shujun Wang analyzed the data, prepared figures and/or tables, and approved the final draft.

Xiaolin Chen conceived and designed the experiments, analyzed the data, authored or reviewed drafts of the article, and approved the final draft.

Zhongyuan Wen performed the experiments, authored or reviewed drafts of the article, and approved the final draft.

The following information was supplied relating to ethical approvals (i.e., approving body and any reference numbers):

The study has been approved by the Clinical Research Ethics Commission of Renmin Hospital of Wuhan University (WDRY2020-K051).

The following information was supplied regarding data availability:

The raw data is available in the Supplemental Files.

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
