# Peer review of "Effect of hyperglycemia on the immune function of COVID-19 patients with type 2 diabetes mellitus: a retrospective study"

_PeerJ, doi:10.7717/peerj.14570_

## Round 0.1 · original submission · Major Revisions

Authors should consider all the issues raised by the reviewers, in particular the post-hoc power calculation for statistical analysis and sample size. Other recent, relevant studies on the same research field should be cited.

·

Basic reporting

INTRODUCTION:
-Regards the role pathogenicity of SARS-CoV2 in humans, and please remark a few relevant concepts as follows:
1. no data has been reported about the cellular expression of ACE2 receptors in endothelial upper ways respiratory cells and humans’ cardiomyocytes. Indeed, it could show the signs of direct SARS-CoV2 intra-myocardial damage (Cardiovasc Diabetol. 2021 May 7;20(1):99. doi: 10.1186/s12933-021-01286-7). In this context, the over-glycation of cardiac ACE2, as showed by post-mortem myocardial samples, could negatively influence the viral entrance and replication in humans, and particularly in hyperglycemics and diabetics (Cardiovasc Diabetol. 2021 May 7;20(1):99. doi: 10.1186/s12933-021-01286-7). Would you please discuss this point and this reference in the text?
2. Again, in the pathogenesis of SARS/COV2 infection, not only the ACE2 receptor but also the serine proteases expression (TMPRSS2) is another relevant point to discuss. Indeed, as reported by authors, its expression in humans’ cells can cause the entrance and replication of SARS/COV2 (miR-98 Regulates TMPRSS2 Expression in Human Endothelial Cells: Key Implications for COVID-19. Biomedicines. 2020 Oct 30;8(11):462. doi: 10.3390/biomedicines8110462). Would you please describe this point and refer to the suggested reference? In my opinion, these informations could improve the Introduction of your article.
3. In this context, we have to say that there could be marked classes of higher risk/worse prognosis patients, for example, hypertensive patients and those with diabetes. Indeed, the hypertensive is at higher risk for COVID19, ICU admission, and deaths (Could anti-hypertensive drug therapy affect the clinical prognosis of hypertensive patients with COVID-19 infection? Data from centers of southern Italy. J Am Heart Assoc. 2020 Jul 7:e016948. doi: 10.1161/JAHA.120.016948). Notably, this study did not evidence any effect of particular classes of anti-hypertensive therapies in COVID-19 prognosis for hypertensive patients. Indeed, it was the first study to evaluate clinical outcomes in hypertensive COVID-19 patients as ACEi vs. ARBs vs. CCBs. Notably, this study has not been evaluated in your submission. Would you please explain this point and the suggested reference?
4. looking at diabetes, you are showing data that have been analyzed and discussed in the last1-2 years. Indeed, I would suggest introducing the chronic hyperglycemic condition (and insulin resistance), as in the case of diabetes mellitus (Outcomes in Patients With Hyperglycemia Affected by COVID-19: Can We Do More on Glycemic Control? Diabetes Care. 2020 Jul;43(7):1408-1415. doi: 10.2337/dc20-0723; Impact of diabetes mellitus on clinical outcomes in patients affected by Covid-19. Cardiovasc Diabetol. 2020 Jun 11;19(1):76. doi: 10.1186/s12933-020-01047-y), but also the hospital admission and acute hyperglycemia as a condition that increases the risk of mortality (Hyperglycaemia on admission to hospital and COVID-19. Diabetologia. 2020 Jul 6:1-2. doi: 10.1007/s00125-020-05216-2; Negative impact of hyperglycemia on tocilizumab therapy in Covid-19 patients. Diabetes and Metabolism 2020; doi: 10.1016/j.diabet.2020.05.005). However, as you could see in the current international literature, the role of diabetes (Diabetes Care. 2020 Jul;43(7):1408-1415. doi: 10.2337/dc20-0723; Impact of diabetes mellitus on clinical outcomes in patients affected by Covid-19. Cardiovasc Diabetol. 2020 Jun 11;19(1):76. doi: 10.1186/s12933-020-01047-y) and hyperglycemia in worse prognosis in COVID-19 (Hyperglycaemia on admission to hospital and COVID-19. Diabetologia. 2020 Jul 6:1-2. doi: 10.1007/s00125-020-05216-2; Negative impact of hyperglycemia on tocilizumab therapy in Covid-19 patients. Diabetes and Metabolism 2020; doi: 10.1016/j.diabet.2020.05.005) has just been previously discussed and published. However, please describe in detail this point. Would you please remark the difference existing between CHRONIC HYPERGLYCEMIA vs. ACUTE HYPERGLYCEMIA? This could improve the quality of the present article. Refer to the suggested reference.
5. Again, there could be an association between the ABO group and clinical outcomes (Implications of AB0 blood group in hypertensive patients with covid-19. BMC Cardiovasc Disord. 2020 Aug 14;20(1):373. doi: 10.1186/s12872-020-01658-z). Indeed, non-0 covid-19 hypertensive patients have significantly higher values of pro-thrombotic indexes, as well as higher rate of cardiac injury and deaths compared to 0 patients (Implications of AB0 blood group in hypertensive patients with covid-19. BMC Cardiovasc Disord. 2020 Aug 14;20(1):373. doi: 10.1186/s12872-020-01658-z). Moreover, AB0 blood type influences worse prognosis in critical patients with covid-19 infection (Implications of AB0 blood group in hypertensive patients with covid-19. BMC Cardiovasc Disord. 2020 Aug 14;20(1):373. doi: 10.1186/s12872-020-01658-z). What is your opinion? Hos is AB0 group represented in your study? I do not see in your article this point. In my opinion this information has to be updated in the text.

Experimental design

METHODS:
You wrote that “The diagnosis of diabetes was based on the 1999 World Health Organization diagnostic criteria: Inclusion criteria: Patients diagnosed with COVID-19 with a history of type 2 diabetes. Exclusion criteria: 1)Type 1 diabetes and specific type diabetes; 2) Pregnancy; 3) History of severe kidney and liver disease; 4) History of autoimmune disease; 5) History of tumor; 6); History of chronic respiratory disease;(7) History of glucocorticoid treatment that “. I do not agree with these diagnostic criteria. Please rewrite all, indicating a valid reference to support it.
How did you collect clinical data? Did you use electronic systems? Who did perform data collection? Who did perform data analysis?
Please indicate how many physicians performed imaging, and the modality to perform the exams. Where they blinded to study cohorts and study protocol or not? Please discuss it.
Did you practice lung echography?
How many cases of acute myocardial infarction did you revise?
A main question. Why did you exclude form the analysis the asymptomatic SARS-CoV2+ patients (ASAP)? Can you analyze the data from this class of subjects as normoglycmeics vs. hyperglycemics vs. diabetics? Indeed, also ASAP are patients with negative prognosis as compared to SARS-CoV2 negative patients (Crit Care. 2021 Jun 24;25(1):217. doi: 10.1186/s13054-021-03643-0). Please respond and discuss it, it is relevant. On the other hand, it is an important study limitation.
How did you calculate sample size of study population?
In Methods report a full descriptive sub-chapter about the laboratory diagnosis of COVID-19 infection.
How did you diagnose and monitor study endpoints? Please discuss it, including all techniques and methods for measuring the study outcomes.

Validity of the findings

RESULTS:
-I would see the study results as:
1. Inflammatory/immune response study results;
2. Human study results (clinical data and outcomes).
Is it possible to show the data in this way? Please discuss it.
-Can you show the effects of hypoglycemic drugs in a “ex-vivo” model to show, during SARS-CoV2 infection the inflammatory/immunological response at baseline and after hypoglycemic drugs?
-In the table 1 there are relevant problems:
1. there are not data about glycemia and insulin resistance in study cohorts. This is not acceptable.
2. there are not data about systolic/diastolic blood pressure, and body mass index of study cohorts.
3. there are not data about medical therapies as anti-diabetic (really frustrating this) and anti-viral (not acceptable). Please correct the table and show the data for further revision process. The anti-diabetics medications and much more the drugs with lowering effects on hyperglycemia could influence clinical outcomes, as suggested before (Diabetes Care. 2020 Jul;43(7):1408-1415. doi: 10.2337/dc20-0723; Diabetologia. 2020 Jul 6:1-2. doi: 10.1007/s00125-020-05216-2; Diabetes and Metabolism 2020; doi: 10.1016/j.diabet.2020.05.005). You have to clarify and discuss these points.
In my opinion the table 1 has not sense. You could include the values of overall in table 1 as separate column with “overall study population” characteristics.
In table 1 include full medical therapy in study cohorts, and the anti-diabetics medications, and the anti-hyperglycemics drugs used.
In the table include the full medical anti-diabetic medication. The table 1 and 2 could be table 1. Please correct it.
Finally, the table 2 does not have any sense. Please update the data in table 1 as unique table. And the same for table 3, 4, 5 that have to be removed and updated as table 1.
Leave the table 6 as NEW-Table 2.
Update a new table with Cox-regression analysis for predictors of worse prognosis.
Can you calculate a Cox regression model to estimate prognostic risk factors for worse prognosis?

Additional comments

It is too long and not well focused on main study results. Please short it and focus it on main study news. Please re-write it as “what is new and what is known”.
In addition, Improve English quality of the text.
Include a study flow chart figure.

Reviewer 2 ·

Basic reporting

Clear, unambiguous, professional English language used throughout = Yes.
Intro & background in context = Yes.
Literature well referenced & relevant = Somewhat outdated, given the circumstances (Aug 2021).
Structure conforms to PeerJ standards, discipline norm, or improved for clarity = Mostly.
Figures are relevant, high quality, well labelled & described. = Yes.

Experimental design

Original primary research within Scope of the journal = Yes, but possibly superceded by research that has already been published.
Research question well defined, relevant & meaningful = Yes, but possibly superceded by research that has already been published.
Rigorous investigation performed to a high technical & ethical standard = Yes, but sample size is small.
Methods described with sufficient detail & information to replicate = Yes.

Validity of the findings

The study is apparently grossly underpowered - no post-hoc power calculations have been supplied by the authors.

Additional comments

Major revision with addition of more subjects and power calculations is suggested.
There are already published studies - have dealt with the issues raised, they need to be cited and discussed.

---

## Round 0.2 · accepted · Accept

The authors have satisfactorily addressed the issues raised by the reviewers.

·

Basic reporting

The manuscript has been improved

Experimental design

The experimental designi is ok

Validity of the findings

The validity of the findings is ok

Additional comments

The manuscript has been improved